# Evaluation of Neutrophil-to-Lymphocyte Ratio (NLR), Platelet-to-Lymphocyte Ratio (PLR) and Systemic Immune–Inflammation Index (SII) as Potential Biomarkers in Patients with Sporadic Medullary Thyroid Cancer (MTC)

**DOI:** 10.3390/jpm13060953

**Published:** 2023-06-05

**Authors:** Roberta Modica, Roberto Minotta, Alessia Liccardi, Giuseppe Cannavale, Elio Benevento, Annamaria Colao

**Affiliations:** 1Department of Clinical Medicine and Surgery, Endocrinology, Diabetology and Andrology Unit, Federico II University of Naples, 80131 Naples, Italy; robertominotta@gmail.com (R.M.); alessia.liccardi@yahoo.com (A.L.); cannavalegiuseppe@yahoo.it (G.C.); elio.benevento@gmail.com (E.B.); colao@unina.it (A.C.); 2UNESCO Chair on Health Education and Sustainable Development, Federico II University of Naples, 80138 Naples, Italy

**Keywords:** neutrophil-to-lymphocyte ratio (NLR), platelet-to-lymphocyte ratio (PLR), systemic immune–inflammation index (SII), medullary thyroid cancer (MTC), inflammation, cancer, biomarker, calcitonin

## Abstract

Medullary thyroid cancer (MTC) is a rare neuroendocrine neoplasm, and calcitonin is its main biomarker. An elevated neutrophil-to-lymphocyte ratio (NLR), platelet-to-lymphocyte ratio (PLR) and systemic immune–inflammation index (SII) have been considered as negative prognostic factors in several neoplasms. The aim of this study is to evaluate the potential role of NLR, PLR and SII as biomarkers in MTC. Clinical data and tumor histological characteristics of patients with sporadic MTC, referred to the NET Unit of Federico II University of Naples (ENETS CoE) from 2012 to 2022, were retrospectively evaluated by analyzing preoperative and postoperative calcitonin, NLR, PLR and SII. We included 35 MTC patients undergoing total thyroidectomy. The mean preoperative NLR was 2.70 (±1.41, 0.93–7.98), the PLR was 121.05 (±41.9, 40.98–227.23) and SII was 597.92 (±345.58, 186.59–1628). We identified a statistically significant difference between pre- and post-thyroidectomy NLR (*p* = 0.02), SII (*p* = 0.02) and calcitonin (*p* = 0.0) values. No association with prognosis or tumor characteristics emerged. Elevated preoperative NLR and SII suggest a possible disease-associated inflammatory response, and their reduction after surgery may be related to debulking effects. Further studies are needed to define the role of NLR, PLR and SII as prognostic markers in MTC.

## 1. Introduction

Medullary thyroid cancer (MTC) is a rare neuroendocrine neoplasm accounting for about 3–5% of thyroid cancers, taking its origin from the parafollicular cells of the thyroid gland [1]. The incidence of MTC is increasing up to 0.21 per 100,000 subjects, without substantial gender differences, and is mainly sporadic [2]. The hereditary form accounts for about 25% of all MTC, shows a genotype–phenotype correlation and is due to RET (REarranged during Transfection) oncogene mutations, with the autosomal-dominant transmission, associated with hereditary multiple endocrine neoplasia syndrome (MEN) type 2A and 2B or the related familial MTC syndrome (FMTC). The disease stage has an impact on prognosis, and the survival rate decreases from 100% in stage I to 21% in stage IV [1]. Notably, recurrent disease occurs in about 50% of patients, and distant metastases are detected already at diagnosis in 10–15% [3]. Great improvements have been realized in MTC therapy with the availability of oral tyrosine kinase inhibitors, vandetanib and cabozantinib, as well as several other options proposed within clinical trials including everolimus and pasireotide and immunotherapies. Nevertheless, the therapeutic algorithm in patients with progressive disease is debated, as well as the timing of starting systemic therapies [4,5,6,7]. Biomarkers in MTC, as in other cancer types, are needed for several purposes including diagnostic screening, the early detection of disease recurrence and progression, the assessment of therapeutic response and prognostic evaluation [8]. Calcitonin is a polypeptide secreted by the parafollicular C-cells of the thyroid gland, encoded by a gene located on the short arm of chromosome 11 [8]. Calcitonin is a widely used sensitive diagnostic and prognostic tumor marker for MTC, having a strong correlation with disease burden [8]. Nevertheless, there are some limitations with calcitonin use including inter-assay variability, rapid degradation and elevation due to other diseases [8,9]. Efforts to identify biomarkers other than calcitonin in MTC have been made, and procalcitonin, chromogranin A, neuron-specific enolase (NSE) and carbohydrate antigen 19-9 (CA 19-9) have been proposed, but this is still an open challenge for clinicians [10]. Additionally, micro-RNAs (miRNAs) have been evaluated as biochemical markers in MTC, similarly to other neuroendocrine neoplasms, though they are not routinely used [10,11,12]. Carcinoembryonic antigen (CEA), an adhesion glycoprotein expressed by neuroendocrine cells, is commonly used in MTC mainly in progressive disease, although it can be elevated also in several other types of cancer and also in many physiopathological conditions including tobacco smoking, gastrointestinal tract inflammatory disease and benign lung disease [11]. Consequently, CEA is not considered a specific marker of MTC, and it is not useful in the early diagnosis of MTC [11]. Inflammation has been related to cancer development and progression, and other factors including endocrine-disrupting chemicals and epigenetic modification are known to interact with inflammation [10,13,14]. The systemic inflammatory response is known to have a role in predicting patients’ survival, not only in neoplastic diseases [15]. It has been demonstrated that platelets can promote angiogenesis and thus the growth of primary tumors, but also evasion by the immune system and metastases development [15]. A pro-tumor action is also supported by neutrophils, releasing cytokines, able to induce immunosuppression, angiogenesis and tumor growth [15]. The role of systemic inflammation in tumor growth and progression has driven attention towards the neutrophil-to-lymphocyte ratio (NLR), platelet-to-lymphocyte ratio (PLR) and systemic immune–inflammation index (SII), an integrated indicator based on peripheral lymphocyte, neutrophil, and platelet counts [16,17]. These are easily measurable, reproducible and inexpensive inflammatory markers and have proven to be reliable in predicting outcomes [16,17]. Previous studies have already identified high NLR values as an unfavorable prognostic marker in COVID-19 patients and in patients with ischemic stroke. The recognition of NLR as an independent predictor of mortality and worse outcome in COVID-19 patients may be helpful to identify high-risk subjects with COVID-19 infection at hospital admission. Similarly, in patients with acute ischemic stroke who underwent endovascular treatment, NLR was related to the occurrence of early neurological deterioration [18,19]. Furthermore, an immunological role of NLR has been proposed also in allergic diseases, finding that NLR may play a key role in regulating the development and exacerbation of atopic dermatitis and allergic asthma [20]. A high NLR ratio has been associated with an increased risk of positive lymph node status in gastric cancer and in non-small cell lung carcinoma (NSCLC), where a high preoperative value of the NLR ratio has been associated with a more advanced tumor stage and a significantly lower 5-year overall survival [21,22]. NLR and PLR ratios also proved to be good prognostic markers in patients with advanced breast cancer treated with neoadjuvant chemotherapy; patients with a low NLR and PLR profile had a higher pathological complete response [23]. SII has been primarily developed based on a retrospective study of patients with hepatocellular carcinoma (HCC) undergoing resection, and it has been demonstrated that SII is a powerful prognostic indicator of poor outcomes in HCC patients [24]. Subsequently, preoperative SII has been investigated in patients with colorectal cancer, and it was considered an independent predictor of overall survival (OS) and disease-free survival [25]. Data about the role of NLR, PLR and SII in MTC are scattered and mainly based on small patient populations, also due to the rarity of the disease, but the available data support an association between inflammation and MTC aggressiveness which deserves a more in-depth analysis [26].

## 2. Materials and Methods

Patients with histologically confirmed MTC referred to the Endocrinology, Diabetology and Andrology Unit of Federico II University of Naples, Center of Excellence (CoE) of European Neuroendocrine Tumor Society (ENETS), from 2012 to 2022, were retrospectively analyzed. Patients with germline mutation of the RET gene, who received any other treatment before surgery, with a history of other active malignancy and known infectious or inflammatory diseases, with a hematological disorder or undergoing immunosuppressive therapy (including corticosteroids) were excluded from the study.

Patients’ demographic and anthropometric data, calcitonin and carcinoembryonic antigen (CEA) levels, histological tumor characteristics, including tumor size, capsular invasion and extrathyroidal extension, stage at diagnosis and recurrence of disease were collected. Biochemically curing the disease was defined as undetectable postoperative serum calcitonin levels during follow-up. The American Joint Committee on Cancer (AJCC) TNM classification of MTC was used to describe the disease stage, according to AJCC criteria [27].

Blood samples including complete blood counts with automated differential counts were taken as part of the clinical routine after overnight fasting and were evaluated if taken within 1 month prior to thyroid surgery and within 3 months after thyroidectomy. Serum calcitonin concentrations were assessed by a commercially available chemiluminescence assay (CLIA), two-site sandwich-type CLIA (DiaSorin Inc., Stillwater, MN). Normal values were 0.4–18.9 pg/mL (male subjects) and 0.0–5.5 pg/mL (female subjects). The preoperative NLR was calculated by dividing the absolute neutrophil count by the absolute lymphocyte count. PLR was calculated by dividing the absolute platelet count by the absolute lymphocyte count. The SII was defined as follows: SII = P x N/L, where P, N and L were the peripheral platelet, neutrophil and lymphocyte counts, respectively. The study was conducted in accordance with the Declaration of Helsinki and approved by the Ethical Committee of the “Federico II” University of Naples (n. 223/19). All subjects signed an informed consent form.

### Statistical Analysis

Statistical analysis was carried out with SPSS version 29.0 software for Windows (SPSS Inc., Chicago, IL, USA). Categorical variables are shown as the absolute number and percentage, while data from non-normally distributed continuous variables (as determined by Gaussian distribution with the Shapiro–Wilk test) are shown as the median and the interquartile range (IQR). Non-parametric data were further analyzed with the Mann–Whitney U test. Correlations between continuous variables were determined by a Wilcoxon signed rank test. A *p*-value <0.05 was considered statistically significant.

## 3. Results

The study cohort included 35 patients, 15 female (42.8%) and 20 male (57.1%), with a median age at MTC diagnosis of 52.1 years (±13.9, range 15–73). MTC diagnosis was histologically confirmed in the entire series, and genetic testing for RET mutation was negative in all patients. The mean follow-up was 62.7 months (m), range 12–78. The main characteristics of our MTC patients’ cohort including their clinicopathological features and disease stage are shown in Table 1.

Regarding the histological characteristics of MTC, the mean tumor size was 19.4 mm (±12.8, range 4–50), seven patients presented capsular invasion (20%) and five patients had extracapsular involvement (14.3%). In our cohort, lymph node metastasis at diagnosis was identified in 17 patients (48.6%), while 2 patients presented with distant metastasis (5.7%), 1 in the liver and 1 in the bone at diagnosis. With regard to anthropometric measurements, 17 patients were normal weight (18.5–24.9 kg/m^2^), 12 were overweight (25–29.9 kg/m^2^) and 6 were class I obese (30–34.5 kg/m^2^). The biochemical evaluation of specific tumor markers included calcitonin and CEA, though the latter was available only in 6 patients. The mean basal calcitonin was 1304.6 pg/mL (range 13–8600) with a modal value of 2000 pg/mL and a normal range between 0 and 18.5 pg/mL (Table 2).

The mean basal CEA was 2813 ng/mL (range 1.6–15,000), with a normal range of <5 ng/mL for non-smokers and <10 ng/mL for smokers (Table 2). The CEA was above the normal reference range in four patients, and among them, one had lymph node and bone metastases and two had lymph node metastases. All patients underwent surgical treatment as first-line therapy: a total thyroidectomy with central neck compartment lymphadenectomy was performed in all patients; in 12 patients (34.3%), a laterocervical lymphadenectomy was also added, and in one case, a hepatic metastasectomy. The pathological disease stage at diagnosis is summarized in Table 1; 19 patients (54.3%) presented lymph node or distant metastases, reflecting the high propensity of this neoplasm to nodal metastatic spread. During the follow-up, disease recurrence was observed in 12 patients (34.2%). The postoperative mean calcitonin value was 650.8 pg/mL (range 1–9560) with a modal value of 1 pg/mL, and the post-operative mean CEA value was 56.3 ng/mL (range 1–1052) with a modal value of 2.7 ng/mL. Only one patient died from CMT during the follow-up due to disease progression. Elevated basal calcitonin values were associated with lymph node and distant metastasis at diagnosis (*p* < 0.01 and *p*: 0.05, respectively), with advanced disease stage (*p* < 0.01) and disease recurrence (*p* < 0.01).

The mean preoperative NLR was 2.70 (±1.41, range 0.93–7.98) (Table 2), and a statistically significant decrease after surgery with a mean postoperative value of 2.18 (±1.16, range 0.48–5.77, Table 2) was observed (*p*: 0.02, Figure 1a). On the contrary, PLR does not show any significant difference before and after surgery: the mean preoperative value was 121.05 (±41.9, range 40.98–227.23) (Table 2) and the mean postoperative value was 122.45 (±49.66, range 15–289.04, Table 2) (*p*: 0.67, Figure 1b). The analysis of SII showed a mean preoperative value of 597.92 (±345.58, range 186.59–1628) and a mean postoperative value of 461.84 (±270.36, range 62.85–1379, Table 2), with a statistically significant decrease, similar to NLR (*p*: 0.02, Figure 1c).

We aimed to assess the prognostic significance of NLR, PLR and SII, and, consequently, we analyzed possible correlations with clinicopathological parameters. Of note, no statistically significant association was observed among preoperative NLR, PLR or SII, or lymph node or distant metastases at the diagnosis, recurrence, or disease stage. Similarly, aggressive tumor histological characteristics, including tumor size, capsular invasion and extrathyroidal extension, were not significantly associated with high preoperative NLR, PLR and SII. Interestingly the statistically significant difference between pre- and post-thyroidectomy of NLR (*p* 0.02) showed the same trend of SII (*p* 0.02) and calcitonin (*p* 0.000), as they all decreased after surgery.

## 4. Discussion

In this study, a monocentric cohort of sporadic patients with histologically confirmed MTC was analyzed to assess the potential role of inflammation-based scores NLR, PLR and SII as biomarkers and their association with clinicopathological characteristics in this rare neoplasm. Several studies have already been published investigating the role of NLR and PLR in many solid neoplasms, including differentiated thyroid cancer, breast cancer and lung cancer, but data about MTC are scant and in a small cohort, albeit recent [16,17,22,28,29]. Growing data suggest that systemic inflammation significantly contributes to tumor development and recurrence through circulating cytokines and chemokines produced by cancer cells. In our study, we included the most common MTC marker, calcitonin, together with the evaluation of new serum inflammation-based scores NLR, PLR and SII, which could gain importance in this field as sensitive markers are of utmost importance in rare neoplasms. CEA was not included in the analysis because preoperative values were available only in six patients, as it is less used as a diagnostic biomarker in comparison with calcitonin and its relevance is mainly related to less differentiated MTC. We aimed to evaluate these inflammation scores as potential biomarkers, making a comparison between pre- and postoperative values to further increase the knowledge of inflammation in MTC and also including an evaluation as potential prognostic factors. Calcitonin is the main biological marker in MTC patients, but, as described in other biomarkers, there are some drawbacks. Indeed falsely elevated calcitonin levels may be detected in some pathological conditions different from MTC, such as multinodular goiter, renal failure, chronic autoimmune thyroiditis, and other thyroid malignancies as well as drugs such as proton pump inhibitors, beta blockers and glucocorticoids [8,30]. Consequently, widely available and inexpensive biomarkers such as NLR, PLR and SII could be useful for clinicians. In this cohort, elevated basal calcitonin levels were detected in all patients. The comparison of NLR before and after the surgical treatment showed a significant reduction. Similarly, a reduction in SII after surgery was detected. Interestingly, these two inflammation markers showed the same reduction as calcitonin values. Calcitonin was evaluated in a study by Xu et al. investigating the potential predictive factors of lymph nodes and distant metastasis in MTC [31]. They found that higher values of NLR and calcitonin concentration were associated with a high risk of metastasis, but no comparison between pre- and postoperative values was provided [31]. In our study, among inflammation biomarkers, only PLR did not show any significant variation. As for calcitonin, we can suppose that decreased disease burden after surgery may have a role in reducing NLR and SII, assuming an influence on the underlying inflammatory process that decreases after the removal of the tumor mass. Interestingly, NLR and SII show the same trend, with a significant decrease for both after surgery. This may also be related to the lack of modification of PLR, and, consequently, SII mirrors NLR.

On the contrary, PLR remained almost stable before and after surgery in our cohort. Platelets are involved in cancer angiogenesis and cell proliferation, migration, and hematogenous metastatic spread due to their interaction with vascular endothelial growth factor (VEGF), platelet-derived growth factor (PDGF), fibroblast growth factor (FGF) and transforming growth factor-β (TGF-β) family proteins [32]. Furthermore, an elevated platelet count was associated with poorer survival in some cancers; nevertheless, the cut-off value for PLR was quite different across studies, and it has been reported that its role could be overestimated [33]. Data regarding PLR in MTC are available in three studies, and in two of them, there is a partial overlap in the populations, and it was found to be an independent predictor of outcomes with higher preoperative PLR related to the presence of lymph node metastasis, higher postoperative recurrence rates and poorer disease-free survival [26,34,35]. A retrospective study enrolled 70 MTC patients (28 males and 42 females) without RET genetic testing with the aim of correlating NLR and PRL with lymph node metastasis and recurrence [34]. Mean pre-operative NLR and PLR were 2.1 ± 0.9 and 112.7 ± 49.4, respectively, and to identify the NLR or PLR prognostic threshold, the authors divided the cohort into two groups using the mean pre-operative values [34]. The low PLR group had a smaller tumor size than the high PLR group, in which there were more metastatic lymph nodes significantly higher in the lateral compartments [34]. A value > 105.3 was demonstrated to be an independent predictor for lymph node metastasis with the best PLR cut-off values in predicting recurrence assessed to be 129.810 or 128.9, 11 levels above which recurrence was more likely [34]. In our study, the lack of modification in PLR may be explained by the relatively small sample size, or a different kind of inflammation could be present in this neoplasm, mainly represented by a neutrophil.

Based on a previous study by Kocer et al. that correlated NLR with recurrence in differentiated thyroid cancer, a higher NLR value, in particular, an NLR >1.9, has been reported to be significantly correlated with multifocal and bilateral disease in MTC patients [34,36]. Nevertheless, NLR did not correlate with MTC recurrence [34,36]. In a study enrolling 78 MTC patients (34 males and 44 females) partly already evaluated in a previous study, with a follow-up of 39.4 months, the aim was to identify which inflammation-based score was more suitable for predicting lymph node metastasis, capsule invasion, advanced tumor stages and recurrence [35]. The pre-operative NLR was 2.2 and the PLR was 112.1. Moreover, the ROC analysis demonstrated that only PLR could be considered as a prognostic factor in MTC patients for predicting lymph node metastasis, capsule invasion and advanced tumor stages, but no statistical significance was observed for disease recurrence. Instead, NLR was not associated with MTC clinical outcomes or recurrence [35]. In contrast, a retrospective study by Xu and colleagues in a population of 61 MTC patients (27 males and 34 females) found a correlation between preoperative NLR and lymph node and distant metastasis, with an NLR > 1.784 threshold, indicating metastases were more likely to occur [31]. Considering the conflicting data, a more recent study evaluated 75 patients (33 males and 42 females) to clarify the role of NLR, PLR and SII scores in predicting the aggressiveness of sporadic MTC [26]. They focused on the histological characteristics and found that high NLR was associated with angioinvasion, extrathyroidal extension and the number of lymph node metastases of the tumor [26]. On the other hand, PLR was exclusively related to extrathyroidal extension, whereas SII was correlated with both the size of the tumor and the number of lymph node metastases [26]. Although these parameters indicate a major biological aggressiveness of the tumor, in agreement with our results, the authors concluded that they have a limited role in predicting the presence or absence of metastases [26]. In our study, the association between NLR, PLR and SII with histological aggressiveness and tumor characteristics, including capsular invasion and extrathyroidal extension, was analyzed, but we did not obtain any statistically significant results in this setting. We could assume that the lack of any association between all the inflammation-based parameters and sporadic MTC prognoses may be due to the relatively good prognoses of our patients or the sample size.

The retrospective nature of this single-center study and the relatively small sample size may represent major limitations. In addition, we cannot rule out the possibility that some patients may have undiagnosed or unreported concomitant illnesses or medications affecting hematopoiesis or systemic inflammation. Some of these limitations are already present in previous studies and are difficult to overcome. Nevertheless, our cohort included only sporadic MTC, and genetic testing was performed in all patients, unlike previous studies, so this could reduce some bias, particularly regarding disease aggressiveness. Moreover, these data also include the evaluation of inflammation scores after surgery, while other studies analyzed only preoperative values. Consequently, this study may offer a more complete evaluation of the inflammation status, shedding new light on the role of NLR, PLR and SII in MTC.

## 5. Conclusions and Future Directions

Inflammation scores including PLR, NLR and SII are inexpensive and widely available biomarkers, whose role in cancer is gaining interest, despite some intrinsic drawbacks mainly represented by low specificity. High NLR, PLR and SII have been associated with poorer outcomes in several cancer types and it has led to large studies in most common neoplasms, but their role in patients with MTC is yet to be defined. Furthermore, whether these scores may predict disease aggressiveness in MTC remains to be demonstrated. The relative rarity of the disease and the often indolent clinical course enhance the need for reliable biochemical markers, even in light of identifying the optimal treatment strategy. Future directions include the measurement of these inflammation markers within the evaluation of MTC patients during the disease course to understand if they could be helpful in the early detection of recurrence. Furthermore, NLR, PLR and SII could be tested before and after target therapies, or rather antiproliferative therapies, as early markers of response. According to these results, NLR and SII could be useful in MTC patients as a high preoperative NLR may reflect a systemic inflammation related to tumor growth. The reduction in NLR and SII after surgery similar to calcitonin may be due to the debulking effects of surgery. Despite previous studies being mainly focused on the prognostic role of inflammation-based biomarkers, they obtained conflicting results. The lack of significant association between NLR, PLR and SII and clinicopathological characteristics in MTC patients currently prevents the use of these inflammation markers as prognostic factors in this setting. Nevertheless, the role of these inflammation markers in other cancer is better defined; thus, they could become a port of the patients’ evaluation in the near future. Further studies with large sample sizes are needed to define serum inflammation scores’ role in the prognosis and follow-up of MTC. Furthermore, the association with other factors impacting inflammation, including obesity, should be taken into consideration in future studies to better understand this association.

## Figures and Tables

**Figure 1 jpm-13-00953-f001:**
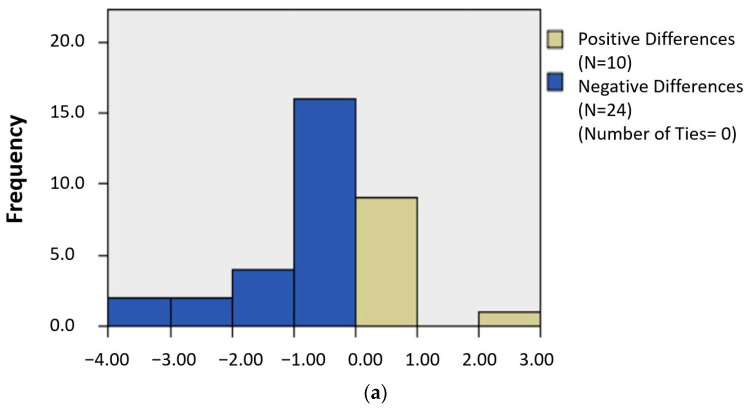
(**a**) related-samples Wilcoxon signed rank test for pre- and postoperative NLR values; (**b**) related-samples Wilcoxon signed rank test for pre- and postoperative PLR values; (**c**) related-samples Wilcoxon signed rank test for pre- and postoperative SII values.

**Table 1 jpm-13-00953-t001:** Baseline characteristics of patients and disease stage according to AJCC 2017 TNM classification of MTC [27].

N tot		35
Gender, F/M		15/20
Age at diagnosis (years), mean ± SD		52.1 ± 13.9
BMI (kg/m^2^), mean ± SD		27.3 ± 5.2
Tumor size (mm), mean ± SD		19.4 ± 12.8
Capsular invasion		7 (20%)
Extrathyroidal extension		5 (14.3%)
Metastasis at diagnosis		
- lymph node		17 (48.6%)
- distant		2 (5.7%)
First-line treatment	Total thyroidectomy + central neck compartment lymphadenectomy	21 (60%)
	Total thyroidectomy + central neck compartment laterocervical + lymphadenectomy	12 (34.3%)
	Total thyroidectomy + central neck compartment laterocervical + liver metastasectomy	1 (2.8%)
Recurrence		12 (34.5%)
Stage		
I		14
II		2
III		7
IV A		7
IV B		2
IV C		2
Follow-up (months), mean		62.7 (12–78)

MTC: medullary thyroid carcinoma; F: female; M: male; BMI: body mass index.

**Table 2 jpm-13-00953-t002:** Pre- and post-operative values of calcitonin, CEA, NLR, PLR and SII.

	Pre-Operative Value ± SD	Post-Operative Value ± SD	
Calcitonin	1304 ± 2219	650.9 ± 1841	*p:* 0.0
CEA ng/mL	2813 ± 6011	56.3 ± 214	*p:* 0.10
NLR	2.70 ± 1.41	2.18 ± 1.16	*p:* 0.02
PLR	121.05 ± 41.9	122.45 ± 49.66	*p:* 0.67
SII	597.92 ± 345.58	461.84 ± 270.36	*p:* 0.02

CEA: carcinoembryonic antigen; NLR: neutrophil-to-lymphocyte ratio; PLR: platelet-to-lymphocyte ratio; SII: systemic immune–inflammation index.

## Data Availability

The data presented in this study are available on request from the corresponding author. The data are not publicly available due to privacy.

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
