# Peer review of "Evaluation of Neutrophil-to-Lymphocyte Ratio (NLR), Platelet-to-Lymphocyte Ratio (PLR) and Systemic Immune–Inflammation Index (SII) as Potential Biomarkers in Patients with Sporadic Medullary Thyroid Cancer (MTC)"

_jpm, 2023, doi:10.3390/jpm13060953_

Round 1

Reviewer 1 Report

Major point

- It would be nice to add the “Conclusions/Future directions ” at the end of the review.

Minor point

- Figure 2-3 should be updated.

Author Response

We thank the Editor and Reviewers for the careful and comprehensive review of our manuscript and valuable advice. In response to the reviewers’ comments and recommendations, we have revised our manuscript and answered all of the questions in a point-by-point manner. All the changes made in the revised manuscript are marked up as requested. We appreciate the reviewers’ efforts and hope that this revised version is now valuable for publication.

Reviewer 1

Major point

- It would be nice to add the “Conclusions/Future directions” at the end of the review.

Thanks for your suggestion, we added “future directions” in the last paragraph.

Minor point

- Figure 2-3 should be updated.

We apologize for the mistake and we have now cited in the manuscript figure 1 a-b-c properly.

Reviewer 2 Report

Here are the comments on the manuscript: 1. the abstract is missing the equal sign at the p-values. 2. the authors should emphasize in the introduction and the discussion what is new in their study. Currently, the result confirming the inflammatory background of the tumor is widely known. 3. the authors should also discuss the role of NLR in other disease entities: doi: 10.3390/life12091415,  doi: 10.3390/ijms22041507, doi: 10.3390/jcm11082235. 4. table 1 and table 2 should be linked. 5. I am asking the authors to explain whether the presence of 2 patients with metastasis may significantly impact the result of NLR and other indicators. It seems that these patients may have had higher NLR and different ratios. 6. a similar remark applies to one patient with liver metastasectomy - did this invasive surgical procedure affect the measurements of inflammatory parameters? 7. statistical analysis is insufficient to conclude - why not more advanced statistical techniques? 8. the discussion is relatively long compared to the rest of the manuscript.

Author Response

We thank the Editor and Reviewers for the careful and comprehensive review of our manuscript and valuable advice. In response to the reviewers’ comments and recommendations, we have revised our manuscript and answered all of the questions in a point-by-point manner. All the changes made in the revised manuscript are marked up as requested. We appreciate the reviewers’ efforts and hope that this revised version is now valuable for publication.

  1. the abstract is missing the equal sign at the p-values.

We apologize for the inaccuracy and we have included the sign in the text.

  1. the authors should emphasize in the introduction and the discussion what is new in their study. Currently, the result confirming the inflammatory background of the tumor is widely known.

Thanks for your advice. We have now emphasized at the end of introduction and in discussion that although inflammatory background has been evaluated in the oncological field, MTC is a rare neoplasm and data about this topic are scarce and in small cohort. Hence, this study contributes to deepen observations about the role of NLR/PLR/SII in MTC.

  1. the authors should also discuss the role of NLR in other disease entities: doi: 10.3390/life12091415, doi: 10.3390/ijms22041507, doi: 10.3390/jcm11082235.

Thank you for your kind advice, we enriched the manuscript, citing these studies in the introduction.

  1. table 1 and table 2 should be linked.

Thanks for your suggestion. In the revised text we linked the two tables as requested.

  1. I am asking the authors to explain whether the presence of 2 patients with metastasis may significantly impact the result of NLR and other indicators. It seems that these patients may have had higher NLR and different ratios.

Probably in the two metastatic patients the advanced disease may have affected the inflammatory parameters, however being our cohort small due to the rarity of MTC, other conclusion or statistical investigations may be unhelpful.

  1. a similar remark applies to one patient with liver metastasectomy - did this invasive surgical procedure affect the measurements of inflammatory parameters?

The single metastatic patients underwent as first line treatment to debulking surgery, without morphological evidence of persistence disease. Hence, probably there was no influence of the metastasis on the inflammatory parameters considered.

  1. statistical analysis is insufficient to conclude - why not more advanced statistical techniques?

In this study we presented preliminary results based on our available data, aware of the limitations mentioned (retrospective nature, single-center study and the relatively small sample size), that unfortunately did not allow more complex static analysis, including subgroup analysis. We plan to increase the number of participants in the near future to perform more advanced statistical techniques.

  1. the discussion is relatively long compared to the rest of the manuscript.

Thank you for the suggestion, our aim was to widely explain the conflicting data from literature. We have deleted some paragraphs to lighten the discussion and to ease reading.

Round 2

Reviewer 2 Report

The answers given by the authors are fully satisfactory.